# Pyrene-Based Fluorescent Porous Organic Polymers for Recognition and Detection of Pesticides

**DOI:** 10.3390/molecules27010126

**Published:** 2021-12-26

**Authors:** Zhuojun Yan, Jinni Liu, Congke Miao, Pinjie Su, Guiyue Zheng, Bo Cui, Tongfei Geng, Jiating Fan, Zhiyi Yu, Naishun Bu, Ye Yuan, Lixin Xia

**Affiliations:** 1College of Chemistry, Liaoning University, Shenyang 110036, China; zjyan@lnu.edu.cn (Z.Y.); ljn1556939707@163.com (J.L.); zhengguiyue1996@163.com (G.Z.); cuibo2019@163.com (B.C.); mgpanda95108@163.com (T.G.); fjt123465789@163.com (J.F.); Yzy20010313@163.com (Z.Y.); 2School of Environmental Science, Liaoning University, Shenyang 110036, China; miaocongke@163.com (C.M.); spj2580@163.com (P.S.); 3Key Laboratory of Polyoxometalate and Reticular Material Chemistry of Ministry of Education, Faculty of Chemistry, Northeast Normal University, Changchun 130024, China; 4Liaoning Key Laboratory of Chemical Additive Synthesis and Separation, Yingkou Institute of Technology, Yingkou 115014, China

**Keywords:** porous organic polymers, pesticides, fluorescence detection, sensor, pyrene group

## Abstract

Eating vegetables with pesticide residues over a long period of time causes serious adverse effects on the human body, such as acute poisoning, chronic poisoning, and endocrine system interference. To achieve the goal of a healthy society, it is an urgent issue to find a simple and effective method to detect organic pesticides. In this work, two fluorescent porous organic polymers, LNU-45 and LNU-47 (abbreviation for Liaoning University), were prepared using π-conjugated dibromopyrene monomer and boronic acid compounds as building units through a Suzuki coupling reaction. Due to the large π-electron delocalization effect, the resulting polymers revealed enhanced fluorescence performance. Significantly, in sharp contrast with the planar π-conjugated polymer framework (LNU-47), the distorted conjugated structure (LNU-45) shows a higher specific surface area and provides a broad interface for analyte interaction, which is helpful to achieve rapid response and detection sensitivity. LNU-45 exhibits strong fluorescence emission at 469 nm after excitation at 365 nm in THF solution, providing strong evidence for its suitability as a luminescent chemosensor for organic pesticides. The fluorescence quenching coefficients of LNU-45 for trifluralin and dicloran were 5710 and 12,000 (LNU-47 sample by ca. 1.98 and 3.38 times), respectively. Therefore, LNU-45 serves as an effective “real-time” sensor for the detection of trifluralin and dicloran with high sensitivity and selectivity.

## 1. Introduction

Pesticide residues have posed huge risks to humans and the ecological environment, causing health problems such as skin irritation, headaches, cancer, and asthma [1]. Pesticides consist of a set of insecticides and herbicides, primarily applied in forestry and crop plantations. In the case of chlorothalonil (CTL), nitrofen (NF), trifluralin (TFL), and dicloran (DCN), these carcinogens accumulate easily in the body, resulting in acute concentrations that cause cancers in humans and livestock [2]. Therefore, various detection methods are being investigated, including capillary electrophoresis [3,4], liquid chromatography [5,6], and electrochemical methods [7,8,9,10,11,12,13]. Although these methods reveal high sensitivity and selectivity, their broad applications are limited by several disadvantages including high cost, complicated operation, and time-consuming pretreatment. The quantitative and visual detection of trace pesticides is still a significant challenge in the field of medico-health and environmental protection.

Fluorescence sensing has the advantages of speediness, convenience, high sensitivity, and “real-time” detection, making it an ideal choice for monitoring toxic pesticides [14]. In recent years, many researchers have reported the use of fluorescence sensors in many applications, such as the detection of trace metal ions, nitro explosives, biological molecules, radioactive iodine, and anions [15,16]. Fluorescent materials play a crucial role in achieving selective and sensitive monitoring. With the development of functional materials, several fluorescent solids, such as carbon nanotubes, carbon dots (CDS), and porous networks, have been widely investigated and have become a newly established sensing system [17]. Among them, porous organic polymers (POPs) attract wide attention owing to their unique characteristics, such as their large surface area, adjustable pore structure, and stable skeleton [18,19]. Based on the advantages mentioned above, POPs have been applied in various fields (electrochemical energy storage and conversion, gas separation, biosensor, catalysis, proton conduction, etc.) and have achieved great success in fluorescent sensor applications [20,21]. Introducing fluorescent chromophores into polymer networks is considered a promising approach to fluorescence sensing detection [22,23]. Pyrene is a polycyclic antiaromatic compound composed of four fused benzene rings. The important features of a long fluorescence lifetime and high fluorescence quantum yield make pyrene a very useful emission chromophore [24,25,26]. During the past few years, more and more work has focused on the construction of pyrene-based POPs [27]. These solids exhibit some excellent characteristics, including good thermal stability, high porosity, and strong luminescence. Based on various synthetic strategies, pyrene-based POP materials with a high density of aromatic fragments provide unique electronic conjugation effects for the enrichment of organic guests [28,29].

Common pesticides, such as trifluralin (TFL) and dicloran (DCN), have electron absorbing nitro or strong electron negative groups (fluorine and chlorine atoms) that can interact with the electron-donating POPs. In this context, two fluorescent porous polymer-bearing pyrene groups are synthesized through the Suzuki coupling reaction. Because of the expansion of the π-conjugated structure, polymer solids enable strong fluorescence performance. LNU-45 has a distorted conjugate skeleton with a high specific surface area and provides a broad interface for analyte interaction. Effective photo-induced electron transfer (PET) is thus produced in the porous structure of POPs after the interaction with pesticides, which is helpful to achieve a rapid response and high detection sensitivity [30]. Therefore, LNU-45 has good selective detection ability for pesticides (trifluralin and dicloran) at ultra-low concentrations, and shows significant potential in the actual detection of organic pesticides.

## 2. Results and Discussion

Two porous polymers (LNU-45 and LNU-47) were synthesized through the Suzuki coupling reaction using 2,7-dibromopyrene (monomer 2) as one monomer together with respective tris(4-boronic acid pinacol ester phenyl)amine (monomer 1) and benzene-1,3,5-triyltriboronic acid pinacol ester (monomer 3) as another monomer (Figure 1a,b). The successful synthesis of the LNU frameworks was determined using Fourier transform infrared (FT-IR) and solid-state ^13^C-NMR spectroscopy. The C-B and B-O stretching vibrations located at ~1360 and 1140 cm^−1^, respectively, were obviously weakened in the IR spectra of the resulting polymers. At the same time, the C-Br stretching vibration at ~760 cm^−1^ for the dibromopyrene monomer disappeared from the FT-IR spectra, verifying the completeness of the Suzuki coupling reaction (Figure 1a,b) [31]. The signal peaks in the range of 120–150 ppm were attributed to the aromatic carbons in the framework, including substituted and unsubstituted aromatic carbons (Figure 1c,d). The carbon atoms of the pyrene group and phenyl carbon atoms bonded to the nitrogen center were observed at 136–140 ppm and 147 ppm, respectively, which confirmed the structural integrity of the LNU networks [32,33].

The thermal stability of the LNUs was characterized by thermo-gravimetric analysis (TGA) under air atmosphere (Appendix A). The skeletal decomposition of the POP solids occurred in the range of 350–400 °C, indicating that the LNUs possessed great thermal stability. Powder X-ray diffraction (PXRD) patterns of the LNUs showed no obvious peaks, which demonstrated the amorphous nature of the polymer networks (Appendix A).

Scanning electron microscopy (SEM) images show that LNUs mainly exist in the form of irregular balls, with an average diameter of 50–100 nm (Appendix A). As illustrated in the transmission electron microscopy (TEM) images (Appendix A), LNUs exhibit a disordered worm-like structure, which demonstrates the porous characteristics of the materials. N_2_ adsorption–desorption isotherms at 77 K and 1 bar reflect the porous nature of the polymers. As shown in Figure 2a, both LNU-45 and LNU-47 have a steep gas adsorption at a relatively low pressure (P/P_0_ < 0.02), indicating the existence of micropores in the POP networks. The specific surface areas of LNU-45 and LNU-47 were obtained using the Brunauer–Emmett–Teller (BET) model to be 322.401 m^2^ g^−1^ and 181.924 m^2^ g^–1^, respectively. The pore size distributions of the LNUs were calculated according to non-local density functional theory (NLDFT), suggesting that LNU-45 possessed micropores at 1.165–1.809 nm and LNU-47 centered at 1.81 nm (Figure 2b). All data for LNU-45 and LNU-47 materials are listed in Table 1.

The solid UV-visible absorption spectra were conducted for LNU-45, LNU-47, and 2,7-dibromopyrene, respectively. (Appendix A). It was observed that the LNUs displayed a strong and wide absorption peak in the region of 250–450 nm compared with the monomer (200–350 nm), because the expansion of the polymeric skeleton increased the π-electron delocalization effect of the conjugated structure [34,35]. Obviously, the peak width of LNU-45 was much wider than that of LNU-47, which was assigned to the π → π* electronic transition of the electron-rich triphenylamine groups in LNU-45.

Under 365 nm UV irradiation, the color of the LNU powders appeared blue (Figure 3a,b). 2 mg of LNU powders were dispersed into 8 mL of methanol (MT), ethanol (EAL), methyl trichloride (TCM), dichloromethane (DCM), tetrahydrofuran (THF), *N,N*’-dimethylformamide (DMF), acetone (DMK), and acetonitrile (ACN). It can be seen from the fluorescence spectrum that there is a small peak at about 650–720 nm, which is attributed to a fundamental peak of the sensing chromophore (Appendix A) [35]. As shown in Figure 3c,f, the LNU samples had the highest fluorescence intensity measured in THF solvent with an excitation wavelength of 365 nm. The fluorescence quantum yields of LNU-45 and LNU-47 were 21.31% and 13.20%, respectively, performing the strongest fluorescence of the materials in THF solvent [36]. Because the enthalpy of π-π interaction between the LNUs and the THF solvent is much larger than that of other solvents, the porous network needs to overcome a high potential barrier to realize the flexible rotation of structural fragment, leading to the higher fluorescence intensity [37].

Several common organic pesticides, including nitrofen (NF), chlorothalonil (CTL), trifluralin (TFL), and dicloran (DCN), were poured into the THF solution of the LNU powders under the same conditions. As shown in Figure 4a,b, the fluorescence quenching was observed as the contact of LNU solids and different organic pesticides compared with the initial dispersion liquid of the LNUs. Notably, both LNU-45 and LNU-47 showed a strong quenching effect on trifluralin and dicloran molecules (Appendix A). Because trifluralin and dicloran contain electron-withdrawing nitro groups as well as strong electron-negative groups (fluorine and chlorine atoms), these electron-deficient molecules attract the energy from the LNU skeleton, leading to the quenching behavior of POP solids [38].

Subsequently, we examined the sensitivity of fluorescent porous organic polymers for the detection of pesticide analytes. The LNUs were evenly dispersed in the same amount of THF solvent, and the pesticide molecules (trifluralin and dicloran) with different concentrations were gradually added into the mixture. With the increase of analyte concentration, the fluorescence quenching effect of the LNUs became more obvious and quenched completely (Figure 5a–d). In order to better explain this phenomenon, X-ray photoelectron spectroscopy (XPS) was used to calculate the relative content of each element (C, N, F, Cl) corresponding to the amount of pesticide adsorbed into the POP architecture [39,40,41,42,43,44,45]. As shown in Appendix A, the adsorption uptakes were calculated to be 116.0 mg g^−1^ (NF-adsorbed LNU-45), 70.0 mg g^−1^ (CTL-adsorbed LNU-45), 139.8 mg g^−1^ (DCN-adsorbed LNU-45), and 231.9 mg g^−1^ (TFL-adsorbed LNU-45). It can be seen that DCN and TFL are more easily enriched in the LNU-45 skeleton, which is conducive to the detection ability of the fluorescence sensor for DCN and TFL at low concentrations [46]. At the same time, the photoinduced electron transfer (PET) occurs between polymer and pesticide molecules. The pesticides containing strong electron-withdrawing groups possess low-energy π*-type orbitals. Their LUMO energies are close to or lower than the LUMO energy of the LNUs, which drive the electron transfer from POPs to pesticides, leading to a strong quenching effect on fluorescence. However, the alkyl group with an electron donating effect will affect the ability of TFL pesticide compounds to accept electron charges [34]. Therefore, LNUs show a strong fluorescence quenching phenomenon with DCN.

The quenching efficiency of the LNUs was calculated by (I_0_-I)/I_0_ × 100%, where I_0_ represents the initial fluorescence intensity and I represents the fluorescence intensity after quenching [47]. The Stern–Volmer mechanism is widely used to describe the relationship between the concentration of a given reagent and the value of quantum yield during photophysical processes. The important parameter *K_sv_* represents a constant of the fluorescence quenching degree caused by the collision between the excited fluorescence molecule and the quenching agent [48]. The sensitivity of the LNUs for pesticide molecules was evaluated by means of the following Stern–Volmer equation:(*I*_0_/*I*) − 1 = *K_sv_* [*C*](1)
where [*C*] is the pesticide concentration, and *K_sv_* is the fluorescence quenching Stern–Volmer parameter [47]. Figure 5e,f illustrate the Stern–Volmer plots of the LNUs for the trifluralin and dicloran, from which the fluorescence quenching Stern–Volmer coefficients are obtained.

A linear fitting was performed to obtain a good correlation between pesticide concentration and intensity. The fluorescence quenching coefficients of LNU-45 and LNU-47 for trifluralin were 5710 and 2880, respectively; the values of LNU-45 and LNU-47 for dicloran were 12,000 and 3550, respectively. This is consistent with the fluorescence intensity obtained by adding different pesticides into the THF solutions of LNU-45 and LNU-47, as shown in Figure 4. The minimum concentrations of trifluralin and dicloran corresponding to the quenching efficiency over 90% for LNU-45 were 9 × 10^−^^4^ and 8 × 10^−^^4^ mol L^−1^, respectively (Figure 5g,h). The limit of detection (LOD) is defined as *3σ/**K_sv_* (*σ* is the standard deviation of the blank solution). The calculated detection limits of DCN were 0.0155 ppm for LNU-45 and 0.0197 ppm for LNU-47, which is better than some reported values, such as 0.13 ppm for BPyTPE, 2.93 ppm for (H_3_O)[Zn_2_L(H_2_O)]·3NMP·6H_2_O, 3.85 ppm for {(Me_2_NH_2_)[In(BDPO)]·DMF·2H_2_O}_n_, 0.212 ppm for 1-L, 0.233 ppm for 3-L, and 0.1753 ppm for DNA-AuNCs [1,14,49,50,51]. The results demonstrated that LNUs could detect trifluralin and dicloran with high selectivity and sensitivity, providing the possibility of detecting residual organic pesticides for practical applications. It is worth noting that the decrease in fluorescence intensity was immediately observed after adding the analyte to the dispersion solution of the LNUs.

More importantly, LNU-45 not only shows high specific recognition for pesticide molecules, but can also be recycled for long-term use. After filtration and activation using ethanol, LNU-45 is able to be used for continuous fluorescence detection of DCN (Appendix A). It is found that LNUs can be reused as sensors for detecting organic pesticides. In addition, the obvious fluorescence quenching phenomenon is observed instantly and intuitively after adding DCN pesticide to the LNU suspensions (Appendix A). Compared with traditional methods, such as capillary electrophoresis, liquid chromatography, or electrochemistry, LNU-45 shows the advantages of simple, fast, high sensitivity, and “real-time” detection in the pesticide detection process.

## 3. Materials and Methods

### 3.1. Materials

Benzene-1,3,5-triyltriboronic acid pinacol ester and tris(4-boronic acid pinacol ester phenyl)amine were purchased from Sukailu, Suzhou, China. 2,7-dibromopyrene, chlorothalonil, nitrofen, trifluralin, and dicloran were all purchased from Aladdin, Shanghai, China. Potassium carbonate was obtained from Energy Chemical, Shanghai, China. Tetrakis(triphenylphosphine)palladium was obtained from Sigma-Aldrich, St. Louis, MO, USA. Other chemicals and solvents were purchased from commercial suppliers and used without further purification. All reactions were performed under a purified nitrogen atmosphere.

### 3.2. Synthesis of LNU-45 and LNU-47

In a 100 mL round bottom flask, tris(4-boronic acid pinacol ester phenyl)amine (400.0 mg, 0.64 mmol) and 2,7-dibromopyrene (346.6 mg, 0.96 mmol) were dissolved in 60 mL of *N,N*’-dimethylformamide. The system was then frozen with liquid nitrogen, with the process of vacuuming and blowing nitrogen repeated three times. Following this process, tetrakis(triphenylphosphine)palladium (40.0 mg, 0.035 mmol) and 5 mL of potassium carbonate solution (2 mol L^−1^) were rapidly mixed into the reaction system. After repeating the above degassing process three times, the mixture was heated to 130 °C and refluxed for 48 h. The precipitates were then filtered to leave the insoluble solids, and washed with tetrahydrofuran, water, and acetone solvents multiple times to remove the unreacted monomers and catalyst residues. The crude product was further purified using Soxhlet extraction with tetrahydrofuran and dichloromethane. Finally, after drying at 90 °C under vacuum for 12 h, the powder obtained was the target product: LNU-45. LNU-47 was synthesized using a similar procedure, except that benzene-1,3,5-triyltriboronic acid pinacol ester (290.9 mg, 0.64 mmol) was used to replace tris(4-boronic acid pinacol ester phenyl)amine.

## 4. Conclusions

We prepared two fluorescent porous organic polymers through the Suzuki coupling reaction using a π-conjugated dibromopyrene building block. Due to the extension of the π-conjugated structure, the generated LNU with a π-electron delocalization effect effectively enhances the fluorescence performance. In contrast to the planar fluorescence structure (LNU-47), the distorted framework provides a broad interface for analyte interaction, which allows LNU-45 to exhibit high sensitivity and selectivity in the “real-time” sensing detection of trifluralin and dicloran. Therefore, the LNU-45 fluorescence sensor has the advantages of being fast, sensitive, and visual, and has broad application prospects in pesticide molecular detection.

## Data Availability

All data related to this study are presented in this publication.

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
