# Peer review of "Pyrene-Based Fluorescent Porous Organic Polymers for Recognition and Detection of Pesticides"

_molecules, 2021, doi:10.3390/molecules27010126_

Round 1

Reviewer 1 Report

The authors have synthesized and characterized luminescent POPs based on pyrene The stress on sensing experiments is poor. The manuscript may be accepted for publication after a revision that attend to the issues pointed out below.

  1. Bands are not clearly visible in the IR spectra. Repeat the experiments and produce better spectra.
  2. Explain the morphology of the polymers from SEM and TEM. Are they spherical/ monolithic, nanopsheres, etc.
  3. References to 13C NMR data?
  4. What are the solid state/solution state quantum yields of both the polymers, i.e. LNU-45 and LNU-47, like?
  5. How about the sensing of pesticides in water or DMSO-water mixtures? Id the polymers do not show fluorescence, why not methanol or ethyl acetate?
  6. The authors mention that LNU-45 and 47 showed stronger interaction for DCN and TFL owing to the presence of electron withdrawing (EWG) group and electronegative groups (EN). However, as can be seen, all the four pesticides, i.e. NF, CTL, TFL and DCN, have both EWG as well as EN groups as part of their skeletons. The reasoning is absurd!

Selectivity should be better explained.

  1. Why does LNU-45 show grater quenching efficiency for all four pesticides compared to that of LNU-47?
  2. What is the concentration in ppb/ppm for sensing? A comparison for the detection limit of pesticides should be made with the other organic fluorescent probes.
  3. Quenching efficiency of POPs by pesticides should be correlated with the LUMO energy of the analyte.
  4. Authors mention that the POPs are applicable for real-time detection. There is no experimental study to show the same!

Reviewer 2 Report

Comments to the Author

This paper is devoted to evolution of designed two fluorescent porous organic polymers LNU-45 and LNU-47 were prepared using π-conjugated dibromopyrene monomer and boronic acid compounds as building units through a Suzuki coupling reaction. Due to the extension of the π-π conjugated system, the resulting polymers with large π-electron delocalization effect revealed enhanced fluorescence performance.  However, I consider that there are some unclear points should be taken into account. These revisions will improve the paper for the publication. The results presented by this work seem interesting, however, this manuscript is not also well organized. In conclusion, this manuscript cannot be accepted in the present form in Molecules before doing a careful revision of the manuscript.

  1. Regarding the abstracts section. The main objective and problem need to be clarified. Moreover, the novelty of the work and the merits of material design need to be further clarified and emphasized. Full names of the abbreviations LNU-45 and LNU-47 should be given in parentheses.
  2. Regarding the introduction part: This introduction part provides good information related to the target and material used in this work, but it is very short and need further clarification, which can be clarified in the following points:

a-, the importance/novelty of the current work needs further clarification

b- More attention should be paid to other efforts made in relation to the materials used in this field

c- Advantages and novelty of the current approach

e- There are some citations that can be used in this review related to the design of materials for other sensor approach for different anlytes detection and applications to ensure subject entirety. It is important for the readers to mention some fluorescent sensor studies and other techniques in which pesticide analyzes are performed to ensure the integrity of the meaning.

Biosensors and Bioelectronics 174 (2021) 112819, Sensors and Actuators: B. Chemical 329 (2021) 129198, Food and Chemical Toxicology 147 (2021) 111886

  1. font size should be slightly larger
  2. As shown in Figure 4c and f, LNU samples had the highest fluorescence intensity measured in THF solvent with an excitation wavelength of 365 nm. Why? It should be explained.
  3. The molecular shapes in figure 5 are not clearly visible. Also, fluorescent spectra should be added to the supporting file.
  4. Error bars should be added to the Fig. 6e-h.
  5. Did the authors also evaluate the response of monomers to these pesticides? What is purpose of polymerization of monomer.
  6. authors should explain the sharp peaks around 650-700 nm. It needs to further clarification.
  7. Sensor mechanism should be added with explanation. Mechanism can support by DFT calculations as following papers: New J. Chem., 2019, 43, 16738-16747.
  8. More details should be added about reproducibility studies.
  9. There are some grammar mistakes this should be also checked in the text.

Reviewer 3 Report

In this manuscript, the authors developed two fluorescent porous organic polymers such as LNU-45 and LNU-47 using π-conjugated dibromopyrene monomer and boronic acids via a Suzuki coupling. Also, the authors claims that these LNU porous organic polymers function as an effective sensors for the detection of trifluralin and dicloran pesticides with high sensitivity and selectivity. These two compounds are characterized by spectroscopic techniques as well as XRD. I believe that this work is useful and may be of interest to other readers of material chemists. Therefore, I am confident that this article will be of the interest for Molecules readers and I will recommend it for publication. There are some minor concerns.

Comments

  1. Is it works for detection of explosives such as TNT, since TNT have three nitro groups.
  2. LNU stands for???
  3. Page 2, line 80 it should be change as Scheme 1 instead of figure 1 and recommend to use numberings for compounds.
  4. Page 2, line 75, 78, etc..: it should be --- (Figure S1, see SI file)
  5. In page 6, line 196, 199, 207.. values (compound weights) should be uniform in all cases.
  6. The coverage of the relevant literature should be improved, e. g. the latest review on C3 Symmetric molecules appeared online in 2019 (https://doi.org/10.1002/asia.201801912)

Reviewer 4 Report

The submitted manuscript describes two fluorescent porous organic polymers LNU-45 and LNU-47 were prepared using π-conjugated dibromopyrene monomer and boronic acid compounds as building units through a Suzuki coupling reaction. Due to the extension of the π-π conjugated system, the resulting polymers with a large π-electron delocalization effect revealed enhanced fluorescence performance. It is determined that the electron transfer from the electron-rich polymer framework to the electron-deficient pesticide molecules causes the quenching phenomenon of the porous materials. Therefore, LNU solids serve as effective "real-time" sensors for the detection of trifluralin and dicloran with high sensitivity and selectivity. This research is well arranged, has a sequence of clear ideas, and concise writing that fits the research plan and methodology. The literature review is good, and they were able to successfully discuss a discussion of their progress, from both a perspective and an applied perspective. Their chosen method makes this data analysis excellent research and enables them to answer research questions and test their hypotheses. As a result of this, I recommend this manuscript for publication in molecules after major revision.

Introduction:

  • Insert a new paragraph to explain the advantages of the Pyrene-based fluorescent porous organic Polymers technique concerning other utilizing techniques?
  • Clarify the benefits of porous organic Polymers make it the desired choice than others?
  • Provide short notes about the applied quenching mechanism photoinduced electron transfer (PET)?

Results and discussion

  • for the fluorescence measurements were detected under the excitation wavelength of 365 nm. What about the scattering peak (is the small peak at around 650 to720 nm)? Is this a scattering, or is this a fundamental peak attributed to the sensing chromophore?
  • Can you provide more information about the Stern-Volmer mechanism?
  • The advantages of this method in comparison with other methods should be highlighted, including analytical characteristics, reproducibility, specificity, stability?
  • No data about reversibility are given?
  • The validation of this technique should be introduced by comparison with a previously validated method?

Round 2

Reviewer 1 Report

I have read all the responses made by the authors. They have addressed most queries. The manuscript seems good after the revision and gives new information about how to detect micropollutants with the help of XPS analysis, mentioned in S7 in SI.  I do think this paper is good for the journal.

Reviewer 2 Report

The authors have improved the manuscript by addressing all the comments/suggestion from the reviewers. Therefore, the manuscript is recommended for publication in its current form.

Reviewer 4 Report

The author made a good modification and reply to my questions successfully. Thus, I agree to accept